# Tests of Dental Properties of Composite Materials Containing Nanohybrid Filler

**DOI:** 10.3390/ma16010348

**Published:** 2022-12-30

**Authors:** Jarosław Zubrzycki, Tomasz Klepka, Magdalena Marchewka, Robert Zubrzycki

**Affiliations:** 1Department of Computerization and Robotization of Production, Lublin University of Technology, 20-618 Lublin, Poland; 2Department of Technology and Polymer Processing, Lublin University of Technology, 20-618 Lublin, Poland; 3Department of Dentistry, Medical University of Lublin, 20-059 Lublin, Poland

**Keywords:** composite materials, nanohybrid filler, dental implants, tribotester, IR spectrum

## Abstract

Complex composite materials are used in many areas of dentistry. Initially, chemically hardened materials were also used, and in this group nanohybrid composites are highly valued. They are often used today, mainly for the direct reconstruction of damaged hard tooth tissue materials for rebuilding damaged tissues using indirect adhesive techniques. The research was conducted to determine the mechanical properties of materials with nanofillers. The article focuses on methods of important test methods for dental prosthetics: resilience, abrasion, wear test, impact strength, hardness, SEM, and chemical analysis. As part of this work, five different series of hybrid composites with nano-fillers were tested. The mechanical properties of composites, such as compressive strength, microhardness, flexural strength, and modulus of elasticity, depend mainly on the type, particle size, and amount of filler introduced. The obtained test results showed that the type and amount of nanofiller have a significant influence on the mechanical and tribological properties. The introduction of nanofillers allowed us to obtain higher mechanical properties compared to classic materials discussed by other researchers. The study observed a change in vibrations in the IR spectrum, which allowed a comparison of the organic structures of the studied preparations.

## 1. Introduction

Many chemical compounds are included in this type of material. These are matrix components, i.e., monomers, e.g., aromatic Bis-GMA resin, and its derivatives, such as Bis-EMA, UDMA, urethane dimethacrylate, carbonates, cyclic esters, acetals, and allyl sulfides. Most of the composite materials, in addition to the essential components, contain monomers such as HEMA, EGDMA, DEGDMA, and TEGDMA, which are lower molecular weight compounds. In addition to the substances mentioned above, composite materials include initiators, activators, and additives required in the polymerization process, e.g., light stabilizers [1,2,3].

Modern composite materials are currently required to be relatively easy to cross-link while maintaining a low polymerization shrinkage value and having unique mechanical and biological properties [4]. Dental composites are a particular type of material that must meet specific requirements for the transmission of loads that work in the conditions of the oral cavity [5]. Therefore, the selection of the chemical composition of the composite is preceded by a thorough analysis of the physicochemical properties of individual components and tests of mechanical properties [6]. The mechanical properties of composites, such as compressive strength, microhardness, flexural strength, and modulus of elasticity, depend mainly on the type, particle size, and amount of filler introduced. The type of coupling agent also plays a vital role in ensuring suitable strength parameters of the composite [7,8].

Wear behavior significantly impacts the mechanical and aesthetic properties of biomaterials. The amount and size of filler particles in composites on the matrix can determine each composite type and, ultimately, the most practical clinical application. It has been reported that damage to composite materials on living tissue can result from degradation of the matrix and filler materials or mechanical and environmental stresses, microcracking, or breakdown of filler particles, which can reduce the survivability of composite restorations in an in vivo setting. In the literature, classifications of composites in some sources focus on properties that determine the viscosity and stability of the composite material, while when considering the mechanical properties of dental composites in other sources, microstructure classifications are taken into account. In the literature, many papers perform two-particle and three-particle abrasion tests on various biomaterials in in vitro laboratory environments [9]. In the contact wear method, while the antagonist material and the composite test material are in direct contact, there is a third structure in the contact range of the non-contact wear mechanism. The mechanical and aesthetic behavior of composite materials, preferred as biomaterials in dentistry, can improve over time. Improvements in the particle size of the dental composite material’s monomer structure have formed the chemical composition of two different particle structures. The filling system of the composite material can be called nano or micro, depending on the particle size. The development of the structure of composite materials has made these materials have excellent mechanical and tribological behavior. In the literature, many studies have investigated the influence of parameters, such as the chewing force, thermal changes, and the wear mechanism of composite materials on mechanical and tribological behavior [10].

## 2. Materials and Methods

The following Figure 1 represents the flowchart that describes the process flow of the experimental study.

Until recently, dental composites had only a classic filler with a particle size of a few micrometers. Currently, composites are increasingly used in which both micro fillers and hybrid nanofillers are introduced. Nanofillers have a particle size from tens of to a few micrometers and tiny particles with a size length of 0.04 µm [11,12]. They are characterized by high mechanical strength (rigid and abrasion-resistant), have a good surface structure, low marginal tightness, and low water absorption compared to composites with micro-fillers [13]. Various materials are suitable fillers for dental composites in the form of spheres, fragments, plates, fibers, or powder. They can be quartz mineral particles, glass (most often barium, strontium, lanthanum, bismuth, etc.), phosphates, silicates, silicon dioxide, and carbon nanotubes [14]. Due to the particle size, fillers can be divided into two main groups: macro filters (particles with dimensions from 0.2 to 70 μm) and nanofillers (particles with dimensions from 0.005 to 0.04 μm). This research aims to show the influence of a hybrid nanofiller’s share on the composite’s selected mechanical properties [15,16].

To prepare the samples for testing, we made casting silicone molds. The tested material was introduced into the mold in layers. Then it was hardened using a specialized UV D-light Duo lamp. The light source was a 5W LED. The wavelength range of the emitted light was 440–480 nm. The maximum power of the lamp was reached at 460 nm. The light intensity of the lamp was 1100 mW/cm^2^. In this way, 25 samples (5 samples for each material) were made in the form of a cylinder with a height of 10 mm and a diameter of 8 mm (Figure 2).

Table 1 provides basic information on the chemical composition of the materials used in the comparative tests.

Charisma Diamond is a composite with a nano-hybrid filler. It is characterized by increased aesthetics, easy polishing of the place where the filling connects with hard tooth tissues, high gloss, low polymerization shrinkage, high flexibility, and optimized hardness [17]. In addition, it has a low viscosity, which makes it easy to model. It can be applied using the layer technique as well as a single layer [18] and has enhanced aesthetic properties for fillings in the anterior and lateral dental arch [19].

Gradia Direct Lo Flo is a micro-hybrid composite made using the technology of pre-polymerized HDR (High Density Radiopaque) fillers, thanks to which it has increased resistance to abrasion. It is characterized by a denser consistency (less liquid) than typical flow composites. It can be used to fill cavities class III and V according to Black’s classification, as well as other classes of cavities, with an emphasis on shallow and small cavities [20]. It also works well as a material for fillings in milk teeth and as a base material. Due to the release of fluorine ions, it has a cariostatic effect.

Charisma Flow is a flow composite characterized by low viscosity and high thixotropic parameters. It is used as a base material, a filling in minimally invasive preparation for shallow and small cavities, and as a fissure sealing material. It has a “chameleon” effect (it can match the color of natural teeth) [21].

Charisma classic is a universal micro-hybrid composite designed to work with the single layer technique, intended for filling cavities in anterior and posterior teeth of all classes, according to Black. Due to the lack of pyrogenic silica in its composition, the inner glow of the material is obtained. Low differences in light refraction between the matrix and the filler have been achieved, as well as the lack of a milky color, which allows for easily selecting the color of the future filling to match the natural tooth tissues. In addition, it comes in a simple range of colors by Vita. The addition of pigment increases the gloss of the color. The opaque colors present in the palette enable better coverage of discoloration in the case of dead, discolored teeth and traces of amalgam filling. Thanks to its creamy and thick consistency, it is easy to apply and model. The composite is easily polished, ensuring high gloss and smoothness of the surface. It is characterized by very low polymerization shrinkage that guarantees long-term marginal tightness, no microleakage, and thus durability, low postoperative hypersensitivity, and a low rate of secondary caries.

Charisma Opal Flow is a flow-type composite with controlled viscosity. It retains its shape and position immediately after application (does not flow from the cavity). It is characterized by low shrinkage tension and high bending resistance. It creates a uniform and smooth surface in hard-to-reach cavities and is applied with a non-drip syringe (no need to retract the plunger after use). It is applied for fissure sealing, as a base material, as a filling in minimally invasive preparations of Black class I, II, III, and V cavities, as a material for splinting teeth, and for minor repairs of direct and indirect prosthetic restorations.

The research program included the following:

1. Research on mechanical properties:Measurements of the coefficient of friction and pin-on-disc friction wear and determination of the degree of wear using the Taylor Hobson profilometer;Impact measurements using a Charpy pendulum hammer;Hardness measurements with the ball indentation method.

2. Tests of chemical composition:Identification of substances, determination of bond types and functional groups, and study of substituent effects in molecules of tested compounds.

## 3. Resilience

The mechanical strength of composites is higher than that of silicon and silicon-phosphorus cement. The cementure power is sufficient to restore the occlusal surfaces of the molars and premolars, as well as the incisors of the front teeth. Reconstructions made of composite materials show high resistance to the oral cavity environment [22,23].

The strength of composite materials consists of their tensile and compressive strength. The tensile strength depends on the matrix’s ductility and the filler particles’ spacing. The highest tensile strength is that of hybrid materials—51.7–66.8 MPa. This value is comparable to the tensile strength of dentin, which is 51.7 MPa. Composite materials with a micro-filter have a tensile strength from 30 to 56 MPa, and composite materials with a macro-filler from 35 to 50 MPa [24]. The tensile strength of composite materials is comparable to that of amalgam, which is 60 MPa [1,23,25,26,27].

The compressive strength of enamel and dentine is 400 MPa and 300 MPa, respectively. Composites with a microparticle filler show low compressive strength—190–260 MPa. Traditional composites are slightly more resistant—250–300 MPa, and the highest compressive strength, comparable to tooth tissues, is that of hybrid composites of 300–450 MPa [2,28,29,30].

## 4. Abrasion

The abrasion resistance of composites intended for anterior tooth fillings is higher than silicon and silica–phosphorus cements. Composite resin consumption is a complex process. On the one hand, it depends on the catalysis and hydrolysis reactions with the participation of enzymes containing saliva, such as hydrolase and esterase. On the other hand, it is related to the physical degradation of the surface. Clinical material wear is manifested by roughness due to abrasion of the matrix, loss of filler particles, fractures of filler particles with the matrix, or exposure to air bubbles.

The abrasion of the material should be comparable to that of the enamel. This is achieved by adding glass, barium, or zinc particles. Too large and complex particles, e.g., quartz, may lead to earlier wear of the enamel of the opposing teeth [31,32].

In the vast majority of preparations with macro filters, the reconstruction area (fillings), despite polishing, isn’t exciting when viewed dry. It turned out that the distance between individual molecules has a decisive influence on the abrasion of composites. Only when this distance is less than 0.1 µm is a tendency to reduce abrasion clear. The average particle distance of the filler in the composite can be reduced by decreasing the particle size of the fill. The use of SiO_2_ filler with a grain size of 0.1–0.5 µm (100–500 Å) significantly reduced the abrasion susceptibility of the composite and made it possible to obtain a mirror gloss composite surface. Thus, composite materials with a micro-filter have superior wear resistance due to a more homogeneous structure and high surface smoothness. The abrasion process uniformly covers the entire material surface, and the small distances between the filler particles create protective conditions for the matrix [33,34].

Traditional macro filter materials show low abrasion resistance due to the abrasion of the resin surrounding the large filler particles, loss of polymer blocks, and fractures between the cohesive filler particles and the matrix.

Materials containing a filler made of micro- and macromolecules, i.e., hybrid composites, show very high wear resistance. They are characterized by slow degradation and increased density and coherence of the organic and inorganic phases. Due to these properties, hybrid materials are the most suitable composites for filling cavities in posterior teeth [11,25,35].

The wear test was carried out in laboratory conditions by measuring the coefficient of friction of a ball (counter-sample) on a rotating disc (sample) pin-on-disc measuring system (Figure 3).

The following measurement parameters were determined experimentally:Probe diameter: 10 mm;Length of the friction path: 2000 m;Load: 3 Mpa;Measurement temperature: 24 °C;Number of measurements on one sample: 3.

The friction coefficient is measured on a pin-on-disk tribotester using the Anton PaarTribo Tester. The average coefficient of friction is measured per one revolution of the sample around its axis.

Figure 4 shows the cumulative graphs of friction coefficient changes during the experiment for the tested samples.

Figure 5 (in mm^2^) and Figure 6 (mass wear in mg) show the wear of the tested materials after tribological tests.

After the wear test, microscopic examinations of the tested samples were carried out. During these tests, the penetration depth and the wear path width were determined. Then, a cross-sectional profile of the friction zone was plotted. Figure 7 shows the three most characteristic microscopic results with the corresponding cross-sectional profile.

The friction coefficient of all tested composites initially increases with the number of cycles and then stabilizes. The highest coefficient of friction, 0.975, was achieved by composite sample 5, and the lowest, 0.769, was acquired by material sample 3.

## 5. Impact Strength

Fracture toughness is measured by the energy needed to cause a material to break. Composite materials with more filler and larger filler particles have higher fracture toughness. For the individual classes of composites, the values of the force needed to break are as follows (from lowest to highest): composite materials with a micro-filler, composite materials with small particles, composite materials with macro-fillers, and hybrid materials.

Fracture resistance in the oral environment is reduced due to water sorption and material degradation.

Impact in terms of mechanics, the resistance of a material to cracking upon impact, is defined. Impact strength is defined as the quotient of the work required to break the specimen and the cross-sectional area at the specimen fracture and is expressed in kJ/m^2^. Toughness is also a measure of the brittleness of a material.

Impact strength of unnotched samples a_n_—Work A_n_ worn for a dynamic fracture of the specimen without a notch relative to 1 cm^2^ of the cross-section at the fracture site:(1)$$\;a_{n}=\frac{A_{n}}{b\cdot h},\left[\frac{J}{{\mathrm{cm}}^{2}}\right]$$
where b—Sample width [cm], h—Sample height [cm].

The impact testing method consists of fists in dynamic braking of the sample and reading the result from the scale of the measuring instrument. The test was performed using the Charpy pendulum hammer.

To determine the impact toughness of plastics using the Izod method, we used a Dynstat-type apparatus, a Charpy swinging hammer, which allowed us to obtain a maximum work in three ranges of 0.5, 1, and 2 J. The hammer blade speed at the moment of hitting the sample should be 2.2 ± 0.05 m/s.

The length of the hammer path varies between 0°–90°, and this value is selected so that the work required to break the sample is within the range of 10–80% of the hammer’s operating range (Figure 8).

## 6. Hardness

The hardness of the material determines its resistance to the forces generated, among others, during chewing. The following factors condition it:

1. Porosity;

2. Type of filler particle;

3. Resin to filler ratio;

4. Properly carried out polymerization polymer.

Depending on the type of composite, the hardness varies. For chemically cured materials, the values are, on average, 15% lower than for light-cured materials (Table 2).

The surface of the composite material is always less challenging and complicated than the deeper layers. This is due to the higher content of organic substances and the inhibitory effect of oxygen on the polymerization process. The surface hardness of the filling can be increased by 2–4% by repeated exposure. Composite materials intended for lateral fillings are characterized by high hardness, similar to amalgam.

Determination of hardness using the ball indentation method consists of slowly pushing the ball into the test sample under the action of a specific load. After a specified time (30 s), the equilibrium state is established, in which the increase in the ng surface of the impression balances the load exerted by the indenting ball. The imprint depth is measured under load. In this state, the ratio of the loading force to the surface of the impression in the material determines its hardness:(2)$$H=\frac{F_{t}}{A},\left[\frac{N}{{\mathrm{mm}}^{2}}\right]$$
where H—ball indentation hardness, F_t_ [N]—applied load, A [mm^2^]—imprint area.

Due to the nature of the deformation during hardness measurements in plastics, hardness is determined by measuring the impression depth “h” under ongoing loading. The ball indentation method only produces reproducible results for specific imprint depths. At small depths, the surface layer of the material may distort the results. At the same time, at significant impressions, the indenting ball only expands the already-formed impression image. The results are calculated from the formula:(3)$$H=\frac{F}{\pi Dh},\left[\frac{N}{{\mathrm{mm}}^{2}}\right]$$
where D [mm]—ball diameter, H [mm]—imprint depth, and F [N]—load.

Table 3 and Figure 9 show the hardness measurement results of individual composites using the ball indentation method.

During the hardness tests, the ball indentation method was used. The highest hardness tested by the ball indentation method, amounting to 302.5 [N/mm^2^], was found in sample no. 3, and the lowest, equal to 183 [N/mm^2^], was observed in model no. 4.

## 7. Chemical Analysis

Identification of the substance, its impurities, the types of bonds and functional groups, and the study of substituent effects in the molecules of the tested compounds were performed using infrared spectroscopy. The FTIR Nicolet 8700 A spectrometer by Thermo Scientific (year of production 2009) was used in the tests. The spectra were recorded by the ATR (Attenuated Total Reflection) method directly from the surface of the samples. IR spectra were recorded in the wavenumber range from 4000 to 400 cm^−1^, with a spectral resolution of 4 cm^−1^. The obtained spectra were subjected to baseline correction operations using the Omnic Specta ™ software for interpretation purposes. Figure 10 shows the cumulative results of the chemical analysis of the tested samples. Figure 11 and Figure 12 show an exemplary IR spectrum for selected samples to more accurately visualize the analysis results of a single material [36,37,38].

## 8. Discussion

Materials used for research contain organic compounds with a complex structure, each having oxygen atoms in the molecule’s structure. Nevertheless, they have a different chemical nature due to the various other ways of incorporating oxygen atoms into their structure: they contain ester, ether, or hydroxyl groups, which translates directly into the physical and mechanical properties of the preparations. Additionally, the presence of secondary amine structures, as well as hydrocarbon chains and rings, influences their properties [39].

The organic compound in sample 5 contains highly hydrophilic hydroxyl groups. Therefore, it shows a greater capacity for water sorption, swelling, and dissolving. This makes the resulting polymer more susceptible to corrosion and the formation of porous structures on the surface [14]. Moreover, as a structure containing aromatic rings, it shows the highest density and viscosity among the tested samples. However, it is characterized by a relatively low rate of weight loss caused by frictional forces, moderate wear, and high hardness [40]. This is due to the ability to create linear polymer structures and three-dimensional cross-links between the chains due to the presence of hydroxyl groups.

Samples 3, 4, and 1 have amine and carbonyl groups in their structure, which significantly affects the polymer hardening process. Under the influence of radiation, carbonyl groups are excited, and the nitrogen atom that forms the amino group is a reducing agent for them, significantly improving the degree of polymerization [41]. Increased efficiency of this process allows for more uniform structures [42].

But the material of sample no. 5 does not contain hydroxyl or amino groups in its molecules; therefore, there are no agents that could support the polymerization process.

The IR spectrum is a graph of the dependence of the absorption of infrared radiation on the frequency of vibrations caused by it in individual structural elements of molecules. It consists of bands (peaks) representing the deformation of the baseline in the form of mountains with a pattern characteristic of a given compound [43]. The spectra analysis is carried out by assigning the type of vibrations occurring in the molecules to the individual bands recorded on the spectrum; it allows for comparing the structures of organic components of the tested preparations. Thanks to the ATR technique, it is also possible to compare the content of individual chemical groups present in the molecules of the components of the tested preparations by determining the ratio of the height of individual unique peaks. The frequencies at which they reached their maximum are given for the most distinct bands in the spectrum [44,45,46,47].

Important information on the differences in the tested substances’ structure is pro- vided by comparing the bands occurring at frequencies higher than 3000 cm^−1^. For samples 3, 4, and 1, the band present in this area is distinct and relatively narrow, a band of stretching vibrations of the amino group. It is single, so a secondary amino group functions in the molecule [48]. For sample 2, the bar in this area is shifted towards higher frequencies; it is also much more blurred; this band corresponds to the stretching vibrations of the hydroxyl group present in the structure of the molecules. Due to lower absorbance than the samples containing amine groups, it can be stated that the -OH groups are much smaller in sample 2 than the =NH groups in preparations 3, 4, and 1. The spectrum of sample 5 does not contain any peak at frequencies higher than 3000 cm^−1^, which excludes the presence of hydroxyl and amino groups in the preparation [49].

The spectrum of each sample contains small bands occurring at the frequency of 2950 –2960 cm^−1^. This proves the presence of fan-shaped methylene groups in the molecule and the presence of aliphatic chains in the structure of the molecules [50]. These bands are the most intense for sample 4, so it is there that these groups are the most numerous. However, the relative height of these vibrations indicates that the alkyl chains in these molecules are very few and short. There are no signs of the presence of aromatic rings for samples 3 and 4; the spectra of samples 2, 5, and 1 suggest the presence of aromatic rings in the structure of the preparation due to a doublet of peaks at frequencies 1608 and 1636 cm^−1^ and, additionally, a clear signal at frequencies around 830 cm^−1^ [51].

All analyzed spectra have an evident, narrow, high band with a frequency of 1715–1725 cm^−1^. It comes from stretching vibrations of the carbonyl group of tested molecules. In addition, the presence of bands of similar intensity at a length of 1030–1050 cm^−1^ derived from -CO stretching vibrations indicates bands of equal power at a distance of 1030–1050 cm^−1^ derived from -CO testing trying beats, indicating the presence of ester bonds in the molecules; they are especially numerous in samples 3 and 5. For the remaining samples, the peak height ratio suggests significant differences in the number of both atom connections. In sample 4, the -CO bonds dominate, which may indicate ether connections [52,53].

Samples 3, 4, and 1 have amine and carbonyl groups in their structure, which significantly affects the polymer hardening process. Under the influence of radiation, carbonyl groups are excited, and the nitrogen atom that forms the amino group is a reducing agent for them, significantly improving the degree of polymerization [54]. Increased efficiency of this process allows for obtaining more uniform structures, which translates into shallow susceptibility to wear and weight loss found in the tests. Sample 3 showed the highest hardness and strength but was characterized by high density and low impact strength. This may result from an inadequate quantitative ratio between the molecules’ carbonyl and amino groups, which directly translates into the monomer binding process in the case of this type of preparation. Samples 4 and 2 showed similar hardness and strength parameter values, but for sample 1, the determined strength and impact toughness were much more favorable. This is probably due to an aromatic ring in the molecules [55].

Sample 5 does not contain hydroxyl or amino groups in its molecules; therefore, there are no agents that could support the polymerization process. As a result, the parameters obtained during tribological tests are unsatisfactory. The highest wear and weight loss among the tested samples result from the insufficiently efficient polymerization process and the formation of a heterogeneous structure. During the study, a favorable impact index was obtained, possibly due to a clear, precise good impact index or aromatic rings and multiple bonds in relatively short molecules. However, relatively low strength and density were obtained [56,57].

## 9. Conclusions

The research shows that using composite materials containing nanohybrid fillers allows for much more durable and aesthetic dental fillings.

Dental composite materials should be used for the direct restoration of damaged dental hard tissues and as restorative and bonding materials for reconstructing damaged tissues using indirect adhesive techniques. The possibility of modifying the composition of both the basic polymer matrix as well as additional components and fillers causes these materials, compared to those previously used, to be distinguished by much better performance properties.

Using nanofillers with particle sizes from 0.005 to 0.04 μm improves the functional and strength properties of dental fillings by about 15% more than materials from the group of macrofillers with particle sizes from 0.2 to 70 μm.

Abrasion and mass wear resistance largely depend on the nanofiller used. It ranges from approx. 3900 μm^2^ up to over 4500 μm^2^ (compare Figure 5)—difference 115.38%.

During the tribological tests, it was found that the friction coefficients increased for all tested materials in the initial phase of the research. After about 11,000 cycles, the values of friction coefficients stabilized. The highest value of the coefficient of friction was observed for sample No. 5—0.975. The lowest coefficient of friction was observed for sample No. 3—0.769 (difference 126.78%).

The type of nanofiller used also significantly affected the hardness of the samples. It ranged from 34.1 MPa to 61.5 MPa in the case of the ball method (a difference of 180.35%). In the Vickers method, the range of changes ranged from 411.7 to 726.7 MPa (a difference of 176.51%).

The chemical analysis showed that the tested materials contain organic compounds of complex structure, each of which has oxygen atoms in the molecular structure. Despite this, they have a different chemical nature due to a different way of including oxygen atoms in their structure: they contain ester, ether, or hydroxyl groups, which translates directly into the physical and mechanical properties of the preparations. Additionally, the presence of secondary amine structures and hydrocarbon chains and rings affects their properties.

## Figures and Tables

**Figure 1 materials-16-00348-f001:**
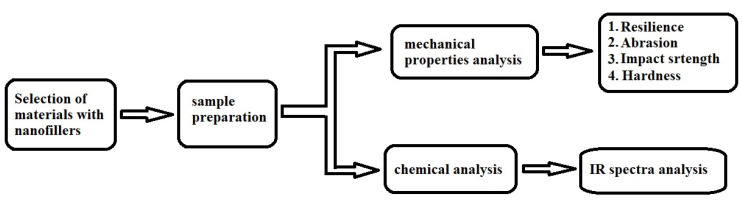
Flowchart of process flow.

**Figure 2 materials-16-00348-f002:**
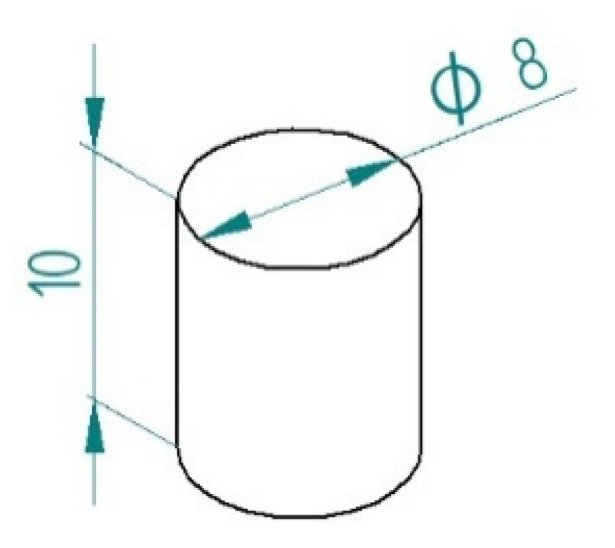
Shape and size of sample.

**Figure 3 materials-16-00348-f003:**
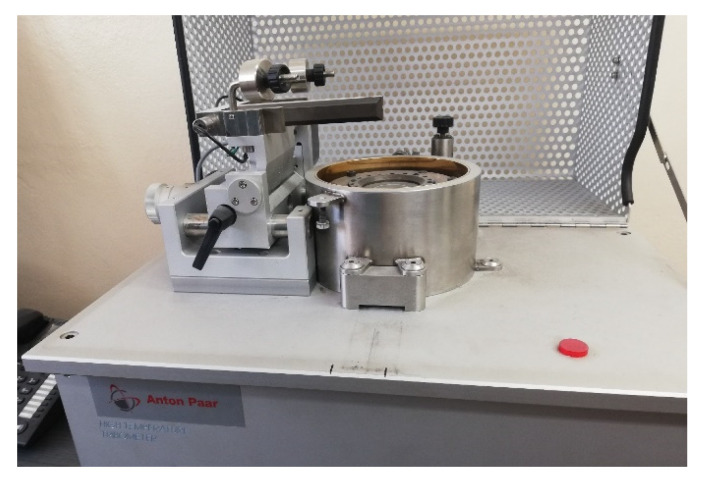
Stand for friction wear tests in laboratory conditions.

**Figure 4 materials-16-00348-f004:**
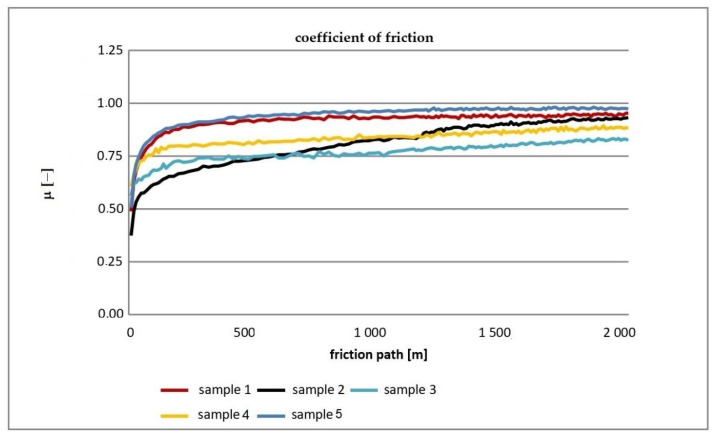
Change of the coefficient of friction for all samples.

**Figure 5 materials-16-00348-f005:**
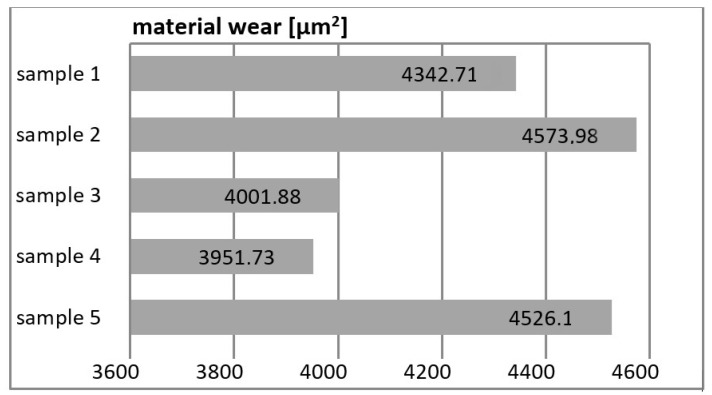
Wear of samples.

**Figure 6 materials-16-00348-f006:**
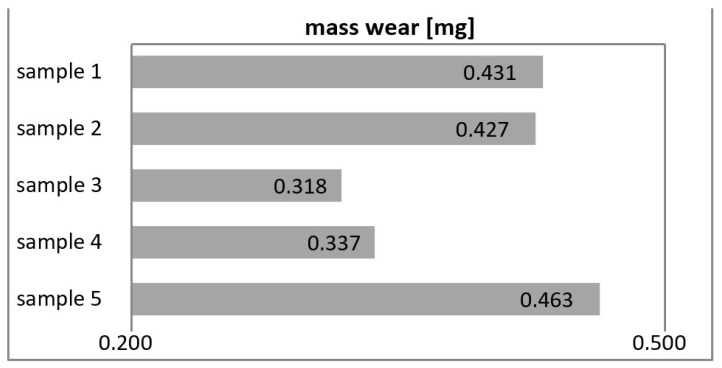
Mass wear of samples.

**Figure 7 materials-16-00348-f007:**
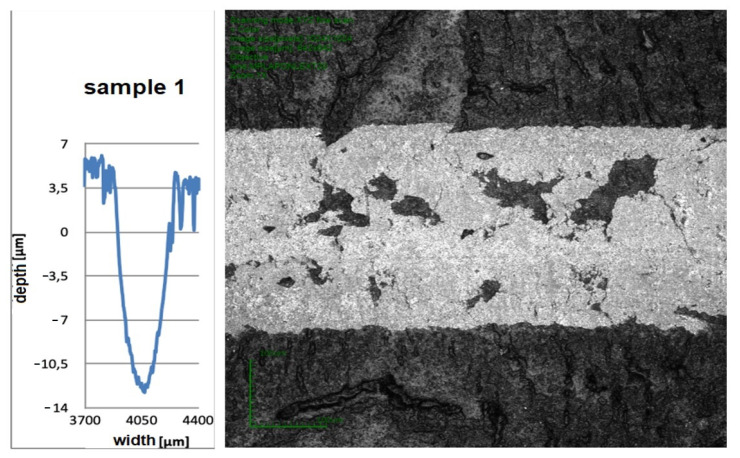
Microscopic examinations.

**Figure 8 materials-16-00348-f008:**
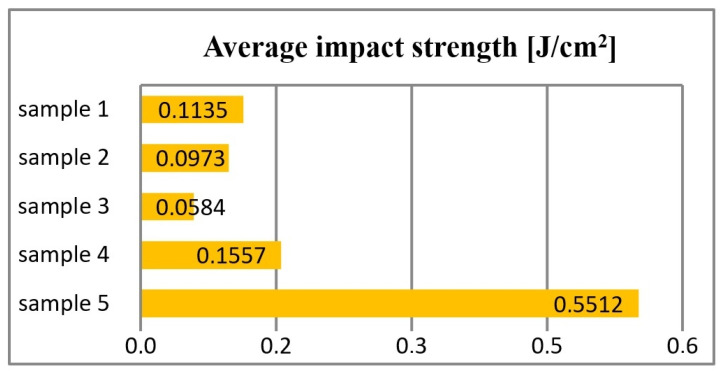
As a result of the impact tests, it was found that sample no. 5 had the highest fracture resistance of 0.5512 [J/cm^2^]. Sample no. 3 was characterized as the lowest impact strength, equal to 0.0584 [J/cm^2^].

**Figure 9 materials-16-00348-f009:**
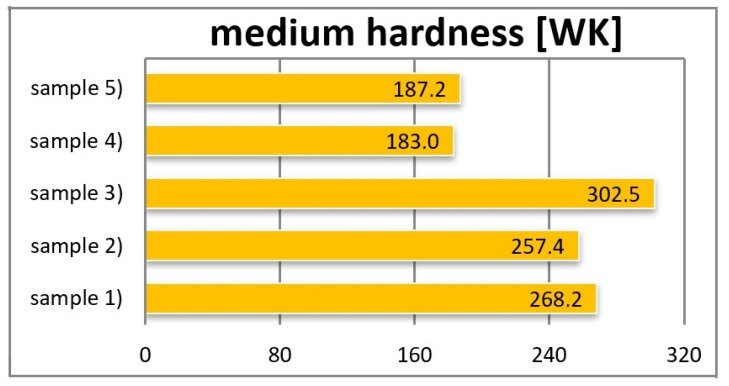
Medium hardness of samples.

**Figure 10 materials-16-00348-f010:**
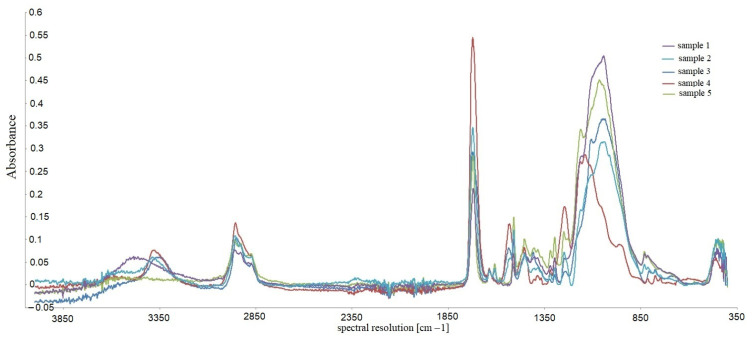
Cumulative IR spectra.

**Figure 11 materials-16-00348-f011:**
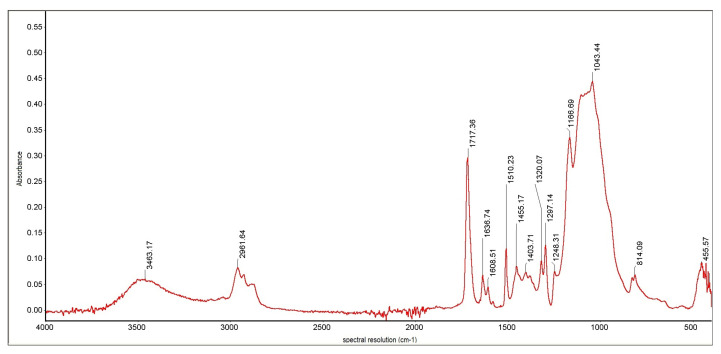
IR spectrum for the sample 1.

**Figure 12 materials-16-00348-f012:**
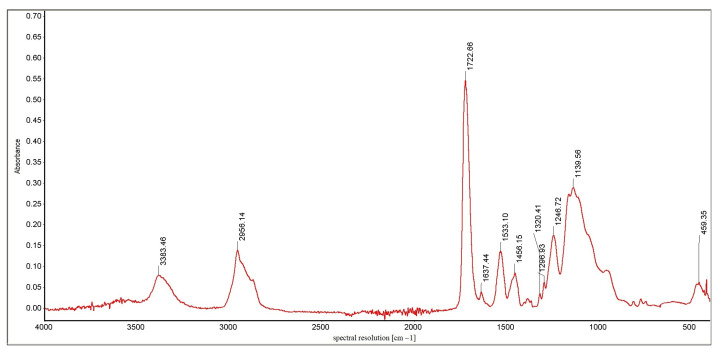
IR spectrum for sample 4.

**Table 1 materials-16-00348-t001:** As part of this work, five different series of hybrid composites with nano-fillers were tested.

NAME	RESIN	FILLERS
Charisma Diamond (1)	-TCD-Di-HEA -UDMA 36%	glass bar- aluminum- fluorine (0.005 μm–0.20 μm) 64%
Gradia Direct Lo Flo (2)	UDMA 37%	-fluoro-alumino-silicon glass -nano-silicon filler -HDR filler 63%
Charisma Flow (3)	-EBADMA -TEGDMA 62%	-silicon school -Ba-Al-F -SiO_2_ 38%
Charisma (4) paste	Bis-GMA 36%	-boron-aluminum-fluorine glass (0.02–0.2 μm) -silicon dioxide (0.02–0.07 μm) 64%
Charisma Opal Flow (5)	-UDMA -EBADMA 39%	-silicon glass nanofiller: Ba-Al, YbF3, SiO_2_ (0.02 μm to 5 μm) 41%

**Table 2 materials-16-00348-t002:** Hardness.

Composite Material	Hardness—Ball Method [MPa]	Hardness—Vickers Method [MPa]
Sample (1)	49.8	417.2
Sample (2)	50.1	431.4
Sample (3)	58.4	560.2
Sample (4)	61.5	726.7
Sample (5)	34.1	411.7

**Table 3 materials-16-00348-t003:** Average hardness results from the ball indentation method.

Composite Material	Wed Hardness [WK]
Sample 1	268.2
Sample 2	257.4
Sample 3	302.5
Sample 4	183.0
Sample 5	187.2

## Data Availability

The data that support the findings of this study are available from the corresponding upon reasonable request.

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
