# Peer review of "Tests of Dental Properties of Composite Materials Containing Nanohybrid Filler"

_materials, 2022, doi:10.3390/ma16010348_

Round 1

Reviewer 1 Report

The topic is relevant to current context but needs to incorporate all the modifications suggested in the below mentioned form

             Abstract looks too much generic! There need to be statistically data to show the significance of the current work. No clarity about the type of material used? What made you to choose this problem? What are all the salient points about this study? What are the limitations of existing materials? Why nano hybrid? What is the benchmark here? What makes them to be compared?

             Title seems to be simple, need to modify to attract more readers

             Abstract should comprise of background, method, results and conclusion.

             Authors quote hydrolysis process except that there are no significant details mentioned in the abstract – to be precise what all tested? With some statistics in broad perspective such as overall enhancement has to be shown in the abstract

             Abstract underscores the content as limitation is 200 words where as currently its 160 words.

             As per journal guidelines 4 to 6 keywords need to be used  

             References should be mentioned as per the journal guidelines

             Authors have to cross verify the quoted references as not more than 2-3 articles are self-cited. Which looks to be bit on higher side (as it’s not a good practice)

             Out of more than 40 papers in reference, only 11 papers are recent which seems to be slightly lower at least more than 50% of the articles should be from recent years such as 2022, 2021 and 2020

·                    Introduction should be focusing on already established methods for various kinds of materials or applications. Few of the recent methodologies can be considered as references –

·                    doi.org/10.3390/polym13172905

·                    doi.org/10.3390/polym13172951

·                    10.1016/B978-0-12-819904-6.00022-0

             There is no information about number of samples considered for testing?

             The entire process could have been explained with process map for easy understanding

             No information about the raw material procurement as well as all the five defined materials

             Why filler sizes restricted to particles with dimensions from 0.2 to 70 μm in case of micro and particles with dimensions from 0.005 to 0.04 μm for nano filler? Justify

             In methods section it is recommended to explain only the process of experiment rather than outcomes (line 77 to line 86)

             All the properties have to be explained with methodology in section 2 and outcome in results and discussion

             There is hardly any discussion with earlier citations from previous works. Why?

             There is no justification for the resulted outcome. Somewhere try to validate and verify the results… why they are behaving so…

             What should be the optimal value based on the predecessor work. That has to be benchmarked and compared with current work

             Figure 9 is not clear and no explanation about it with peer article comparison

             There need to be standard deviation or tolerance value …

             Any characterization study such as FT-IR, EDX, SEM and TEM would have helped in detail to justify the results.

            Any statistical/simulation tool usage to reduce the experimental work or to reduce the number of parameters which are non-critical or significant while testing.

            Best practice while mentioning the graphs to maintain a uniformity of units (SI unit usually preferred)

            Results and discussions are acceptable only with any correlation between the current used material and existing materials in public domain

            There is no comparative study with analytical/experimental/simulation for validation of the extracted results

             There is no discussion section in the manuscript. Without the discussion section how, any conclusion is accepted

             Conclusion section is lengthy, part of it can be considered under discussion section

             The discussion section should have good no. of article comparison to arrive at outcome comparison.  

             The work seems limited and doesn’t have validation or verification with any other statistical/simulation work is missing?

             Conclusion looks to be generic need to compile the outcomes and state based on the tests conducted and convey how best this can fit in the current context for any application.

             In conclusion section, values have to be displayed with explanation. It's better to mention the salient features of the entire work in terms of bullet points with current context

Author Response

Dear Reviewer, thank you for your thorough review. We send our responses in the attached file.

Best regards

Authors

Reviewer 2 Report

In the article entitled “Tests of dental properties of composite materials containing nanohybrid filler” the authors investigated mechanical properties of nanofillers materials used in prosthetic dentistry. The article in is line with journal’s aim, and the Authors have well revised several issues; however, I ask authors to add some key concepts.

-       The abstract section is confusing (basic fundamental?), the authors need to break it down in the following order:

Background,

Aim (describe the null hypothesis),

Materials and methods,

Results,

Conclusions

-       The introduction section is too short, the authors must implement the bibliography and improve the contents presented, also discussing other materials such as carbon nanotubes, used to improve the mechanical characteristics of composites (please see and discuss DOI: 10.1002 / chem. 201302704)

-       In the part of materials and methods the authors must only describe the mechanical tests performed, in the part of the discussion instead they must discuss the results obtained and compare them with when the current state of the art

-       The section related to scanning electron microscope (SEM) analyzes requires references on the use of the protocol to evaluate the fracture topography of the tested material (please see and discuss DOI 10.3390 / ma13133026)

-       It would be appropriate to include a flow chart of the study and some laboratory images on sample preparation

-       What are the limits of the study?

-       Conclusions cannot be reduced to a sentence: you must improve them highlighting the limits and the future insights pointed out from this article.

-       The formatting of the references is not correct, please check the journal instructions for authors

-       Several moderate typos are present in the text, please, amend

According to this Reviewer’s consideration, novelty and quality of the paper, publication of the present manuscript is recommended after major revision.

Author Response

Dear Reviewer,

thank you for your insightful review. The comments contained in the review were reflected in the major revision paper. All changes are highlighted in yellow. The paper after major revision is added to the attachment.
Sincerely yours

Authors

Author Response

(The authors gave the same response as above.)

Round 2

Reviewer 1 Report

The authors have addressed the comments of reviewer successfully. I recommend this manuscript for acceptance in its present form.

Author Response

Dear Reviewer,
thank you for your fruitful cooperation
Authors

Author Response

Dear Reviewer,

Thank you for your insightful review and valuable comments that allowed us to improve our work. Text fragments added, based on your comments, are marked in green in the updated version of the paper. Thank you for your fruitful cooperation
